# The perception of individuals with low back pain regarding reassuring information: Insights based on physiotherapists messages

Ron Shavit[1], Talma Kushnir[2,3], Yaniv Nudelman[4,5], Shmuel Springer[1]*

1 The Neuromuscular & Human Performance Laboratory, Department of Physical Therapy, Faculty of Health Sciences, Ariel University, Ariel, Israel, 2 Department of Psychology, Ariel University, Ariel, Israel, 3 Adelson School of Medicine, Ariel University, Ariel, Israel, 4 Department of Physical Therapy, Ariel University, Ariel, Israel, 5 Maccabi Healthcare Services, Tel-Aviv, Israel

* Shmuels@ariel.ac.il

## Abstract

### Background

Clinical guidelines for the management of low back pain (LBP) emphasize the importance of reassuring patients, as this reduces concern and increases confidence in recovery. Although physiotherapists (PTs) often use reassurance strategies, the perception of the different reassuring messages remains unclear.

### Objective

To investigate perceptions of confidence of different reassurance messages delivered by PTs to people with LBP.

### Methods

A survey was conducted among 544 participants with LBP. The survey included 21 reassuring messages divided into six themes 'Prevalence and statistics', 'Red flag', 'Natural healing', 'Imaging', 'Treatment strategies', and 'Pain physiology'. Participants had to rate the extent to which each message increased their confidence using a 5-point Likert scale. Messages were categorized into four levels of perceived confidence based on participants' responses: high (≥80% rated the message as 'provide confidence' or 'significantly provide confidence'), moderate (60–79%), low (40–59%) and very low (≤40%). Correlation and non-parametric analyses examined the relationships between confidence ratings, demographic variables and personality traits.

### Results

Messages that emphasized patient autonomy and the absence of red flags were perceived as particularly reassuring and received the highest ratings (≥80%). In contrast,

**Data availability statement:** An anonymized data set is availability at the following link: https://doi.org/10.6084/m9.figshare.29896376.

**Funding:** The author(s) received no specific funding for this work.

**Competing interests:** The authors have declared that no competing interests exist.

messages that referred to natural healing and the neurophysiology of pain were perceived as less effective in providing confidence. Personality traits and background characteristics had minimal effects on perceived reassurance.

## Conclusions

Reassuring communication that emphasizes patient autonomy and the low likelihood of serious pathology can boost confidence in individuals with LBP and may be more impactful than patient-specific characteristics. These findings may help PTs refine their communication strategies and strengthen therapeutic relationships, potentially leading to better treatment outcomes. Future research should explore the implications of these findings in clinical settings in real-life interactions.

---

## Introduction

Physiotherapists (PTs) play a critical role in the assessment and treatment of low back pain (LBP), a common and often debilitating condition that affects millions of people worldwide [1,2]. Persistent LBP can significantly affect not only physical functioning, but also psychological well-being, quality of life, family dynamics, social interactions and work commitments [3]. Consequently, effective treatment of LBP requires a multidimensional biopsychosocial approach that encompasses the complex interplay of biomedical, physiological and psychosocial factors [4].

In line with best practice, guidelines for the management of patients with LBP emphasize the need to reassure patients [5]. The process of providing reassurance in PT involves both affective and cognitive components, such as relationship building, generic emotional reassurance, and cognitive reassurance [6]. Building relationships focuses on fostering trust, with the clinician showing genuine interest in the patient's condition and empathy for their concerns. Generic reassurance is about emotional support through reassuring messages such as 'you do not need to worry' or 'everything is fine' emphasizing that there are no serious health concerns. Cognitive reassurance is about giving the patient clear information about their condition and diagnosis, explaining to them how the proposed treatment will solve their problem and ensuring that they understand this explanation [6]. Pincus et al. [7] found that cognitive, affective and generic reassurance play a crucial role in meeting patients' needs. It was also found that reassurance should not only target anxiety but also encourage physical activity to reinforce positive coping mechanisms and thus reduce the risk of chronicity [8].

Reassurance is a fundamental component of LBP treatment, particularly in primary care, where it serves to alleviate patient anxiety, promote self-efficacy and prevent overutilization of healthcare resources [5]. Minz et al. [9] have shown that people with LBP are often afraid of a serious illness or a misdiagnosis. Such concerns may lead patients to seek additional consultations or unnecessary imaging procedures, emphasizing the need for clinicians to proactively address these concerns during consultations. Simonsen et al. [10] further emphasized the importance of cognitive

reassurance, demonstrating its association with better outcomes, such as reduced pain and disability in individuals with LBP. This suggests that clinicians should not only rule out serious pathology but also provide clear explanations that help patients reinterpret their pain in a less threatening way. Traeger et al. [11] underscored the means of shifting patients' perspectives from fear and avoidance to acceptance and proactive management, and reported that patients with acute or subacute LBP felt more reassured and less anxious when they were given clear information about the low likelihood of serious pathology. The reduced utilization of healthcare resources after patient reassurance suggests that this is an effective strategy to control costs associated with non-specific LBP [12,13].

To enhance reassurance, clinicians should provide clear, logical explanations of the medical condition and treatment plan to increase understanding and confidence in recovery. An effective approach should also empower patients' autonomy, i.e., their ability to make informed decisions about their care by actively involving them in their care through shared decision making and evidence-based explanations of their condition, emphasizing the typically favorable prognosis [14,15]. Eliminating common misconceptions, such as incidental imaging findings that are often unrelated with symptoms, can further reduce anxiety [16]. In addition, patients can be empowered with self-management strategies, tailored rehabilitation plans and education on how to recognize warning signs [17].

Nevertheless, the reassurance approach is not without its critics. Some argue that reassurance, if overgeneralized or insufficiently tailored, can trivialize a patient's experience or fail to address specific fears [18]. Darlow et al. [18] point out that patients with chronic pain often need more than generic reassurance to feel truly understood, especially if the reassurance does not address specific fears related to movement, prognosis, or recurrence. Furthermore, it has been suggested that reassurance can inadvertently focus patients on symptom monitoring rather than recovery [19], which can be described as a paradox where health-related anxiety actually increases [19]. Cashin et al. [20] caution that reassurance that focuses solely on symptom relief without addressing the psychological dimensions may inadvertently promote avoidance behaviors, ultimately undermining the benefits of physical activity and engagement.

PTs commonly attempt to reassure patients by emphasizing the absence of red flags [21]. They also educate patients on safety netting, which involves structured plans for symptom management, addressing fears of serious conditions, and advising when further investigation is needed [11,22]. Using this strategy, PTs inform patients about the typical course of LBP, educate them about symptoms within the normal range, advise them when further investigation is indicated [23,24], and empower them to take an active role in their recovery [25,26]. However, the impact of these strategies on patient perceptions is still unclear. Furthermore, screening for red flags may inadvertently increase patient concern [21,27], as differing beliefs, perceptions and health literacy may lead patients to misinterpret findings as indicative of serious pathology [7].

Although existing research provides some insight into patients' perceptions of reassuring messages from PTs [28,29], there are still significant gaps in understanding whether and how patients interpret these messages to increase confidence that their condition is not dangerous. Better understanding of patient perceptions could help PTs refine communication strategies, foster stronger therapeutic relationships and improve treatment outcomes.

Perceptions of reassurance messages may be influenced by traits associated with neuroticism, such as emotional instability, anxiety, catastrophizing, and self-consciousness [30,31]. These characteristics, combined with prolonged symptom duration, can lead to skepticism, anxiety, misinterpretation of reassurance messages and resistance to reassurance [32–34]. It is therefore valuable to investigate whether longer symptom duration, together with higher levels of emotional instability, anxiety, catastrophizing and self-consciousness, influences patients' perceptions of reassuring communication.

The aim of this study was to investigate the confidence perception of reassuring messages, as conveyed by PTs, in people with LBP. We also examined the influence of personal and clinical characteristics on these perceptions.

## Methods

A purpose designed survey was carried out to investigate whether and to what extent people with LBP have greater confidence in the safety of their condition in response to various reassuring messages. The study involved three phases: (1)

an expert panel collected data on reassuring messages commonly used by PTs and designed an initial survey, (2) a pilot testing evaluated and refined the survey, and (3) a large-scale survey was conducted to analyze individuals' perceptions of reassuring messages. The study followed established guidelines [35–37] and is reported in accordance with STROBE [38] and CHERRIES guidelines [39]. Ethical approval (AU-HEA-SS-20230125) was obtained from Ariel University, and all participants gave informed consent by accessing the survey landing page, which included details of the study and ethical information.

## The survey development

To ensure face and content validity [40,41], we assembled an expert panel of PTs with 10–32 years of clinical experience in treating patients with LBP and varying qualifications: 14 bachelor's degrees, 10 master's degrees, and 3 PhDs. Each PT documented messages they commonly used to reassure patients with LBP. This resulted in 97 messages that reflect real-life therapeutic communication. Messages were synthesized to eliminate redundancy, and thematic analysis was conducted. Thematic analysis was chosen over other methods (such as grounded theory) as the aim of the study was to identify and organize patterns in the data without developing a new theoretical framework, which is consistent with the exploratory aims of this study [42,43]. Two investigators (RS and SS) performed independent initial coding, followed by a collaborative analysis to establish consensus on the synthesized messages and emerging themes. The analysis process followed an iterative approach in which messages or themes were continually refined by merging, modifying, expanding, or redefining them. To improve analytical rigor, the coding framework underwent a comprehensive review by the entire research team. The thematic analysis revealed 21 different main messages and six themes: *'Reassurance based on back pain prevalence and statistics' ('Prevalence and statistics'), 'Reassurance based on red flags clearance' ('Red flags'), 'Reassurance based on natural healing of back pain and positive recovery expectations' ('Natural healing'), Reassurance based on interpretation of imaging results' ('Imaging'), 'Reassurance based on treatment strategies' ('Treatment strategies'),* and *'Reassurance based on explanation of pain neurophysiology' ('Pain physiology').* Messages were randomized to prevent grouping effects and order-related response bias [44,45]. The instructions for the participants were: *"Imagine that your PT gives you the following message about your current condition during a session. For each message, rate the extent to which it increases your confidence and removes fears or doubts about pain and illness".* Each message was rated on a 5-point Likert scale: 1 – significantly decrease confidence, 2 – decrease confidence, 3 – neither decrease nor provide confidence, 4 – provide confidence, 5 – significantly provide confidence. A professional language editor has reviewed the final survey to improve its readability, consistency, formatting, and linguistic clarity. The final survey with the 21 messages according to the six themes is presented in S1 Appendix.

## Pilot testing

During pilot testing, 34 individuals with LBP completed the survey and were interviewed about clarity, relevance, completion time, and possible improvements. Three participants indicated that message no. 3, a reassuring message about the absence of findings suggestive of cauda equina syndrome, was somewhat unclear due to overly professional wording. Consequently, the wording of this message was slightly revised. No other issues were reported.

## The large-scale survey

The survey was launched online, and participants were recruited via social media in support groups for people with LBP, with permission from the administrators to share the link for the survey. This approach increased the representativeness of the study and is considered reliable for social media surveys [46,47]. The sample size was calculated using the 95% margin of error formula, aiming for an estimated margin of error of 4.5% with a target size of 500 participants [48]. Prior to starting the survey, participants had to provide written informed consent by responding to a mandatory consent question. They were informed that they could withdraw from the survey at any time by simply closing the survey window. The survey

was completely anonymous and no identifying information was collected from participants. The survey data was collected from October to December 2024. Inclusion criteria included Hebrew language fluency, age between 18 and 70 years, and self-reported LBP, regardless of the duration of pain. Background characteristics were also collected to identify potential confounders such as age, gender, pain duration, education level, numerical pain rating, recent or current PT treatment, consultation with a spine specialist, spinal imaging, and neuroticism, which was assessed using the Hebrew-translated neuroticism subscale of the Big Five Inventory (BFI) [49], a validated and widely used personality assessment tool [50,51]. The Neuroticism subscale comprises 8 items that can reliability assess the person degree of emotional stability or instability [52], three of which are reverse-coded to ensure a balanced assessment of neuroticism and emotional stability.

## Data analysis

Since confidence perception was measured on an ordinal scale (1–5), we analyzed frequency distributions and used descriptive statistics to illustrate participants' perceptions of the different reassuring messages. A message rated above 3 indicated that participants perceived it with at least some level of confidence (i.e., 'provide confidence' or 'significantly provide confidence'). To identify the messages that inspired greater confidence, we categorized them into four levels based on response distribution: high confidence (≥80% of participants rated above 3), moderate confidence (60–79% rated above 3), low confidence (40–59% rated above 3), and very low confidence (≤40% rated above 3). The score for each theme was calculated by averaging the corresponding responses for each participant. As the aggregated scores for all six themes were not normally distributed, they are reported using the median and the interquartile range (IQR). Differences in the distribution of scores between the six themes were analyzed using Friedman's test and post-hoc pairwise comparisons were performed using the Bonferroni correction. To examine the relationships between participants' characteristics and the perceived confidence ratings of messages and themes, Spearman's rank correlation coefficient was used to assess associations with continuous variables (age, symptom duration, pain rating, BFI personality scores), as the messages' scale was ordinal, and the data was not normally distributed. Kruskal-Wallis and Mann-Whitney U tests followed by Bonferroni corrected pairwise comparison were used to assess differences in messages and themes ratings across categorical demographic variables (gender and educational level) and clinical history (previous PT experience, imaging studies, and specialist consultations). Effect sizes (Cohen's r) and correlation coefficients were categorized as small (0.1), moderate (0.3), and large (0.5) [53,54]. All statistical analyses were performed using SPSS (version 28, IBM Corp., Armonk, NY), with statistical significance set at $p < 0.05$.

## Results

### Participants' characteristics

A total of 544 individuals who self-reported experiencing LBP took part in the survey. The majority were female (54.8%) and had a high level of education, 78.1% had either a bachelor's degree (46.7%) or a master's degree (31.4%). Regarding seeking medical care for LBP, 61% of respondents consulted a specialist, while 33.8% had received PT treatment in the past 6 months, and 16.5% were currently undergoing treatment. Table 1 summarizes the characteristics of the participants.

### Perceived confidence across messages

The analysis revealed varying perceived confidence levels (i.e., the percentage of participants rating above 3) across messages. The messages categorized as perceived with high confidence (≥80% cumulative responses rated above 3) included *'Patient autonomy'* (0.88, 95% CI: 0.85–0.91), *'No signs of cancer'* (0.85, 95% CI: 0.82–0.88), *'No concerning signs found'* (0.83, 95% CI: 0.79–0.86), *'No signs of infection'* (0.81, 95% CI: 0.78–0.84), and *'Gradual activity ↓ pain'* (0.80, 95% CI: 0.76–0.83). Among these, *'Patient autonomy'* had the highest cumulative percentage (87.9%), indicating substantial confidence perception among respondents.

**Table 1. Participants characteristics (N = 544).**

| Age (years) | 45.4 ± 12.8 |
|---|---|
| Symptoms duration | |
| One to six weeks | 206 (37.9%) |
| Six to twelve weeks | 40 (7.4%) |
| Above twelve weeks | 298 (54.8%) |
| Female | 298 (54.8%) |
| Education Level | |
| High School | 119 (21.9%) |
| Bachelor's Degree | 254 (46.7%) |
| Master's Degree | 171 (31.4) |
| PT Treatment in Last 6 Months | 184 (33.8%) |
| Currently Receiving PT Treatment | 90 (16.5%) |
| Consulted Specialist | 332 (61%) |
| Imaging Type | |
| None | 262 (48.2%) |
| MRI | 105 (19.3%) |
| CT | 97 (17.8%) |
| X-ray | 71 (13.1%) |
| PET-CT | 9 (1.7%) |
| Neuroticism total score, mean (SD) | 20.01 (5.99) |

PT = Physical Therapy; MRI=Magnetic Resonance Imaging; CT = Computed Tomography; PET-CT = Positron Emission Tomography and Computed Tomography

The messages categorized as perceived with moderate confidence included *'No signs of cauda equina'* (0.77, 95% CI: 0.73–0.80), *'Serious causes are rare'* (0.76, 95% CI: 0.72–0.80), *'No signs of disc herniation'* (0.74, 95% CI: 0.70–0.78), *'No signs of fracture'* (0.72, 95% CI: 0.68–0.76), *'Disc herniations natural healing'* (0.69, 95% CI: 0.65–0.73), *'Beneficial treatment options'* (0.67, 95% CI: 0.63–0.71), *'Safety netting'* (0.65, 95% CI: 0.61–0.69), and *'Pain resolves with time'* (0.63, 95% CI: 0.59–0.67).

The messages categorized as perceived with low confidence included *'No need for special treatment'* (0.57, 95% CI: 0.53–0.61), *'Age-related changes'* (0.57, 95% CI: 0.53–0.61), *'Common experience'* (0.44, 95% CI: 0.39–0.48), *'Pain intensity ≠ severity'* (0.45, 95% CI: 0.41–0.49), and *'Imaging not needed'* (0.41, 95% CI: 0.36–0.45).

The messages perceived with the lowest confidence included *'Common findings ≠ pain'* (0.34, 95% CI: 0.30–0.38), *'Pain is multifactorial'* (0.33, 95% CI: 0.29–0.37) and *'Imaging findings ≠ pain'* (0.27, 95% CI: 0.23–0.31). These messages also had a higher proportion of responses rated below 3 (26.5%, 15.3%, and 26.2%, respectively), indicating potential negative perceptions among participants. Fig 1 illustrates the distribution of the percentage of participants who rated the various reassurance messages as above 3.

### Theme ratings and comparative analysis

Analysis of theme ratings revealed significant differences in perceived confidence across most themes, as identified by the Friedman test (test statistic = 755.649, df = 5, p < 0.001). Table 2 presents the post-hoc comparative analysis between themes. As can be depicted in Table 2, The two highest rated themes were *'Red flags'* and *'Treatment strategies'*, both with a median score of 4.00 and a low interquartile range (IQR = 0.67), indicating a high and consistent level. However,

 

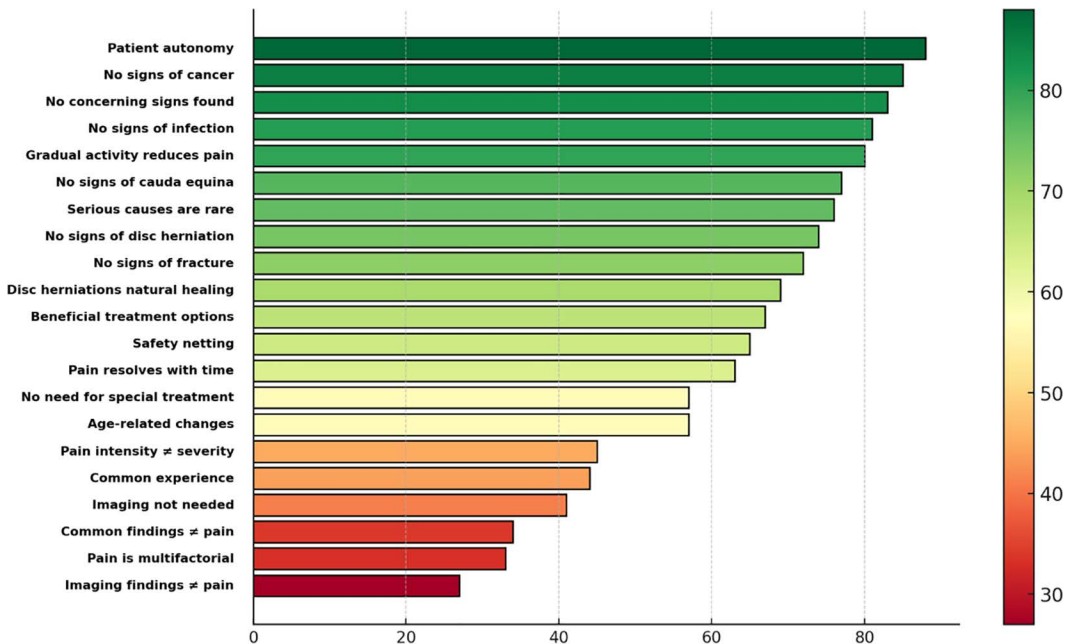

**Fig 1. The percentage of participants rating above 3 ('provide confidence' or 'significantly provide confidence') of the reassurance messages.**

**Table 2. Themes comparative analysis.**

| Theme (Median, IQR) | *Prevalence and statistics* (P values) | *Red flags* (P values) | *Natural healing* (P values) | *Imaging* (P values) | *Treatment strategies* (P values) | *Pain physiology* (P values) |
|---|---|---|---|---|---|---|
| *Prevalence and statistics* (4.00, 1.50) | -- | -- | -- | -- | -- | -- |
| *Red flags* (4.00, 0.67) | <0.001 | -- | -- | -- | -- | -- |
| *Natural healing* (3.75, 0.94) | 1 | <0.001 | -- | -- | -- | -- |
| *Imaging* (3.25, 1.00) | <0.001 | <0.001 | <0.001 | -- | -- | -- |
| *Treatment strategies* (4.00, 0.67) | <0.001 | 1 | <0.001 | <0.001 | -- | -- |
| *Pain physiology* (3.00, 1.00) | <0.001 | <0.001 | <0.001 | 1 | <0.001 | -- |

IQR = Interquartile Range; *Prevalence and statistics* = Reassuring using prevalence and statistics; *Red flags* = Reassuring using red flags clearance; *Natural healing* = Reassurance based on natural healing of back pain and positive recovery expectations; *Imaging* = Reassurance based on interpretation of imaging results; *Treatment strategies* = Reassurance based on treatment strategies; *Pain physiology* = Reassurance based on explanation of pain neurophysiology

no differences (p = 1.000) were found the comparisons between *'Natural healing'* and *'Prevalence and statistics',* between *'Treatment strategies'* and *'Red flags'* and between *'Imaging'* and *'Pain physiology'*. Fig 2 presents the median confidence scores across the reassurance themes.

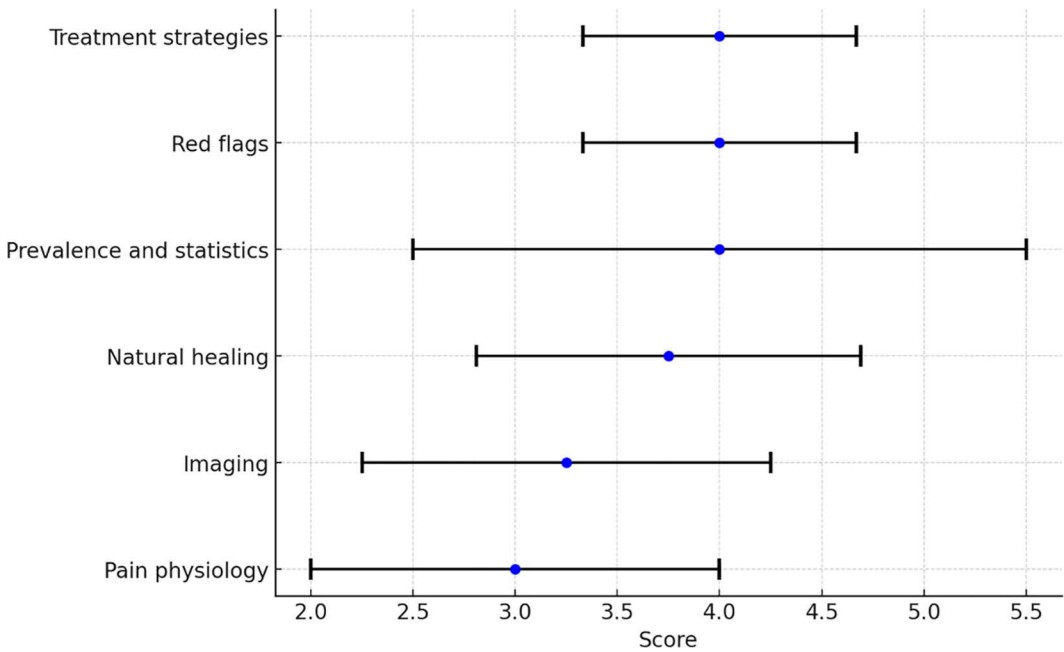

**Fig 2. The median confidence scores across the reassurance themes with error bars representing interquartile ranges.**

### The relationships between participants' characteristics and perceived confidence

The association analysis between background variables, patient personality characteristics and perceived confidence in the messages revealed several significant correlations, though of low magnitude (Spearman's rho < 0.2). No significant differences in message ratings were found across categorical variables.

Low correlations were also found at the theme level, with the exception of the theme *'Natural healing'* which correlated with symptom duration with Spearman's rho = −0.21 (p < 0.001). In addition, significant differences in theme ratings were found between the different education levels and imaging categories. However, the effect sizes were very small, with the exception of the theme *'Natural healing'*. In particular, people who had undergone MRI rated this theme higher than people who had not undergone imaging. This difference was statistically significant (Z = 4.905, p < 0.001, Cohen's r = 0.21), suggesting that undergoing advanced imaging was associated with higher confidence in natural recovery. No differences were found in relation to previous or current PT, specialist consultation or gender. These results are summarized in S2 Supplementary File.

### Discussion

This study provides important insights into how people with LBP perceive reassuring messages from PTs and improves our understanding of patient communication in musculoskeletal care. While previous research has examined the general role of reassurance in clinical interactions [8,55], this study explores the specific content of messages used by PTs in everyday practice and their perceived impact on patient confidence. Through a structured process, real-life reassurance messages from experienced clinicians were collected, refined, and thematically categorized into six major themes to represent the key strategies PTs use in their efforts to reassure patients with LBP. By analyzing the content and perceived impact of these clinical messages, the study provides a practical and relevant framework that can support the refinement of reassuring strategies in routine PT encounters.

Based on the response distribution, our analysis showed that the *'Patient autonomy'* message from the *'Treatment strategies'* theme was particularly effective in providing confidence, with 87.9% positive perception, while the other message in this theme (*Beneficial treatment options*) had a moderate response with 66.6% positive perception. These results aligns with research suggesting that patient empowerment can promote confidence and active involvement in recovery [17], and should underscore the importance of integrating autonomy into treatment-related communication strategies.

Consistent with previous research emphasizing the critical role of education in reducing anxiety and promoting patient confidence [11], messages related to the absence of red flags, such as *'No signs of cancer', 'No concerning signs found', and 'No signs of infection'*, were also perceived with high confidence. This suggests that clear, explicit communication through reflection and explanations to reassure patients about the low likelihood of serious pathology could be effective in alleviating concerns. Interestingly, the message "*No signs of fracture'* elicited only moderate responses, possibly indicating that patients generally may not perceive spontaneous fractures as a major concern associated with LBP. Another possible explanation for this is that fractures are often seen as conditions that can heal, whereas diagnoses such as cancer, infections or neurological impairments are perceived as more serious and less resolvable and can therefore cause greater anxiety [56]. PTs may apply this insight by addressing patients' specific fears and misconceptions in their communication.

While also related to patient education, messages about the natural healing of LBP were generally perceived as offering low to moderate reassurance. This may be due to their non-specific nature, which can make them less impactful in increasing confidence. Notably, the most effective message in this category was '*No signs of disc herniation'*, which explicitly clarifies why it is not a herniated disk and achieved a confidence rating of 74%, further supporting this notion. In addition, a weak to moderate negative correlation was found between ratings of the *'Natural healing'* theme and duration of symptoms, with individuals with long-lasting symptoms reporting lower perceived confidence in these messages, possibly reflecting increasing skepticism over time. These findings highlight the need for specific and tailored cognitive reassurance, especially for those with persistent symptoms, consistent with Darlow et al. [18], who noted that non-specific reassurance is often inadequate in patients with chronic pain.

Consistent Weisman et al. [46], who reported that people with pain often hold negative attitudes toward neurophysiological explanations, messages related to *'Pain physiology'* and the interpretation of imaging techniques were perceived as less effective in providing confidence, with many falling below the 40% positive perception threshold. This diminished perception may also stem from the messages being implicit rather than directly reassuring, and from the use of overly technical language that is less accessible to patients. It is also possible that messages related to *'Imaging'* theme might inadvertently introduce a degree of uncertainty, as they can highlight ambiguities or limitations in diagnostic imaging rather than providing clear, reassuring conclusions. This uncertainty may, in turn, undermine the ability to effectively provide reassurance, potentially leaving patients with lingering doubts about their condition [57]. Despite their limitations, these messages might play a supportive role when integrated into broader therapeutic interactions. Future research should explore how implicit reassurance messages can be framed more effectively to ensure they are meaningful to patients and address both the cognitive and emotional dimensions of their concerns.

Respondents who had undergone MRI found the *'Natural healing'* messages more reassuring than those who had not. However, given the small effect size (Cohen's r = 0.21) and prior research indicating that MRI scans may have an iatrogenic effect, increasing anxiety or reinforcing maladaptive beliefs [58], this result should be interpreted with caution. Although additional correlations were found between personal/background characteristics to the responses to reassuring messages and themes, they were very weak. This suggests that the content and delivery of messages are likely to have a greater effect than personality and background characteristics. It is possible that the effectiveness of reassurance strategies is more strongly influenced by the ability to address patients' specific fears and concerns than by demographic or psychological factors. Therefore, an effective and direct communication approach, such as asking about the patient's specific fears, can help to tailor reassurance to the patient's individual concerns and thus improve the therapeutic relationship.

Although this study provides insight into the potential impact of reassuring messages by PTs, some limitations should be noted. The use of a survey based on written messages did not take into account non-verbal cues such as tone of voice and facial expressions, which are central to conveying empathy and building trust [55], and it may also led to interpretation bias as participants may have understood the messages differently than intended [59]. While this limits the ecological validity to some extent, it allowed us to isolate the content of the reassuring messages and can assist in designing written and digital communication tools beyond face-to-face interactions. Future research should therefore investigate these reassuring messages in real-time face-to-face interactions, including verbal and nonverbal communication, to gain a more comprehensive understanding of reassurance strategies. Secondly, recruiting participants via social media may limit generalizability, as these platforms often attract individuals more engaged in professional communities or more proficient with technology, leading to selection bias [60]. To mitigate this issue, we used a relatively large sample. In addition, our study was aimed at people with LBP. However, it is possible that people with different clinical presentations, such as radicular symptoms, may require a different type of reassurance. Finally, the expert panel designing the survey included only PTs, and participants were asked to imagine their PT delivering the reassurance messages. Message credibility varies by provider, with reassurance from physicians often perceived as more authoritative, particularly regarding serious pathology [11]. This may limit the generalizability of our findings to other healthcare settings. However, with the increasing direct access to PT services, PTs are often the first point of contact for musculoskeletal complaints, which make reassurance a critical part of their role.

## Conclusions

Reassurance strategies that emphasize patient autonomy and absence of red flags were found to be perceived as more effective in increasing the confidence among individuals with LBP compared to messages based on prevalence and statistics, natural healing, imaging and pain physiology. Personality traits and background characteristics were not associated with the perceived impact of reassurance messages, suggesting that well-structured, clear, and direct communication may be beneficial across individuals, regardless of differences in personal disposition or health history.

## Supporting information

**S1 Appendix. Final survey.**
(DOCX)

**S2 Supplementary file. Additional results.**
(DOCX)

## Acknowledgments

We acknowledge the contribution of all participants who gave their time and provided valuable insights into this research.

## Author contributions

**Conceptualization:** Ron Shavit, Talma Kushnir, Shmuel Springer.

**Data curation:** Ron Shavit.

**Formal analysis:** Ron Shavit, Yaniv Nudelman.

**Investigation:** Ron Shavit, Shmuel Springer.

**Methodology:** Ron Shavit, Shmuel Springer.

**Project administration:** Ron Shavit, Shmuel Springer.

**Software:** Yaniv Nudelman.

**Supervision:** Talma Kushnir, Shmuel Springer.

**Validation:** Ron Shavit, Yaniv Nudelman, Shmuel Springer.

**Writing – original draft:** Ron Shavit.

**Writing – review & editing:** Ron Shavit, Talma Kushnir, Yaniv Nudelman, Shmuel Springer.

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
