## [Decision Letter · Decision Letter 0]

27 Jun 2025

Dear Dr. Springer,

Thank you for submitting your manuscript to PLOS ONE. After careful consideration, we feel that it has merit but does not fully meet PLOS ONE’s publication criteria as it currently stands. Therefore, we invite you to submit a revised version of the manuscript that addresses the points raised during the review process.

 The reviewer's comments are available below. Could you please carefully revise the manuscript to address all comments?

We look forward to receiving your revised manuscript.

Kind regards,

Steve Zimmerman, PhD

Senior Editor, PLOS One

Journal Requirements:

2. Peer review at PLOS ONE is not double-blinded (https://journals.plos.org/plosone/s/editorial-and-peer-review-process). For this reason, authors should include in the revised manuscript all the information removed for blind review.

Reviewers' comments:

Reviewer's Responses to Questions

**Comments to the Author**

1. Is the manuscript technically sound, and do the data support the conclusions?

Reviewer #1: Yes

2. Has the statistical analysis been performed appropriately and rigorously?

Reviewer #1: Yes

3. Have the authors made all data underlying the findings in their manuscript fully available?

Reviewer #1: Yes

4. Is the manuscript presented in an intelligible fashion and written in standard English?

Reviewer #1: Yes

Reviewer #1: INTRODUCTION - well written. Some comments below.

1. lines 32-38 - would suggest using the phrase biopsychosocial approach to come in line with the language being used in research and clinical practice. The authors are essentially saying this in their writing, just would make it explicit.

2. lines 56-63 - here it appears there is one referenced statement after another. Thank you for the amount of research and references provided, it is important. However, I am curious if the authors can expand a bit in between those three referenced sentences adding how that may relate to clinical practice. Just a suggestion please.

3. Overall, the introduction is longer than I have typically seen in more recent manuscripts. However, for this particular paper, I think it is completely justified.

METHODS

1. Line 134 - thematic analysis was used. Why did the authors choose thematic analysis rather than grounded theory for this study? I am not challenging, I am asking as I think readers would want to know as not everyone is familiar with these terms.

2. Lines 174-175 - inclusion criteria - self reported LBP, any LBP with or without radicular symptoms? This to me will make a difference when you are studying reassurance. Please clarify.

3. I did not see a discussion of sample size for this survey. Can the authors provide some guidance or thoughts on this and if this was considered to be able to provide a response rate?

4. Statistics are solid and clearly articulated with the objective identified for each analysis run.

RESULTS

Excellent summary of the results with appropriate tables and figures.

DISCUSSION

1. lines 285-287 - I think this first sentence needs to have stronger language as it is critical to start the discussion. The phrase "shed light" while appropriate is not strong enough language. Please revise. I say this because the next sentence has the evidence to back up stronger language.

2. In the limitations section, I would ask the authors to consider that in this study it appears that any back back was include whether their was radicular symptoms or not. I think this is a limitation as different reassuring strategies may be needed with different types of LBP.

Overall, I would like to thank the authors for this important manuscript that addresses a critical area of management in individuals with LBP.

**Do you want your identity to be public for this peer review?** For information about this choice, including consent withdrawal, please see our Privacy Policy

Reviewer #1: No

---

## [Author Response · Author response to Decision Letter 1]

9 Jul 2025

Response to reviewer’s comments on manuscript PONE-D-25-18786

We thank the editor and the reviewer for the opportunity to consider a revised version of our manuscript. We have revised the paper and incorporated the reviewer's suggestions. Below, we summarize each of the comments raised by the reviewer and present our responses. Changes are highlighted in yellow.

Journal Requirements:

Response: We have carefully revised our manuscript to ensure that it meets PLOS ONE's style requirements, including file naming conventions, as requested.

2. Peer review at PLOS ONE is not double-blinded (https://journals.plos.org/plosone/s/editorial-and-peer-review-process). For this reason, authors should include in the revised manuscript all the information removed for blind review.

Response: As requested, we have reinserted all information that was previously removed for blind review into the revised manuscript.

3. We note that you have indicated that there are restrictions to data sharing for this study. PLOS only allows data to be available upon request if there are legal or ethical restrictions on sharing data publicly

Response: There are no legal or ethical restrictions on sharing the data from our study. We have uploaded the anonymized data set necessary to replicate our findings to a stable, public repository at the following link: https://lifesciences.datastations.nl/dataset.xhtml?persistentId=doi:10.17026/LS/PZJ0K5

4. Your ethics statement should only appear in the Methods section of your manuscript. If your ethics statement is written in any section besides the Methods, please move it to the Methods section and delete it from any other section. Please ensure that your ethics statement is included in your manuscript, as the ethics statement entered into the online submission form will not be published alongside your manuscript

Response: We have moved the ethics statement to the Methods section and removed it from all other sections of the manuscript, as requested.

Response: We have added captions for all Supporting Information files at the end of the manuscript and have updated the in-text citations accordingly to match PLOS ONE’s guidelines.

Response: We have reviewed our reference list to ensure that it is complete and correct. No retracted papers are cited in our manuscript, and no issues were identified.

Response: We have processed all figure files using the PACE tool and re-uploaded them after ensuring they meet PLOS requirements.

Reviewer

1. Lines 32-38 - would suggest using the phrase biopsychosocial approach to come in line with the language being used in research and clinical practice. The authors are essentially saying this in their writing, just would make it explicit.

Response: Thank you for your comment. We have revised the text to explicitly use the phrase “biopsychosocial approach” to align with current research and clinical practice terminology, as suggested. Please see line 43 of the revised manuscript.

2. Lines 56-63 - here it appears there is one referenced statement after another. Thank you for the amount of research and references provided, it is important. However, I am curious if the authors can expand a bit in between those three referenced sentences adding how that may relate to clinical practice. Just a suggestion please.

Response: We have expanded this section to clarify how these findings relate to clinical practice, as suggested. Please see lines 63-66 and 68-70.

3. Overall, the introduction is longer than I have typically seen in more recent manuscripts. However, for this particular paper, I think it is completely justified.

Response: Thank you for your comment and for your understanding regarding the length of the introduction. We appreciate your feedback and are glad to hear that the length is justified for this manuscript.

4. Line 134 - thematic analysis was used. Why did the authors choose thematic analysis rather than grounded theory for this study? I am not challenging, I am asking as I think readers would want to know as not everyone is familiar with these terms.

Response: Thank you for your comment. We opted for thematic analysis because our aim was to identify and organize patterns in the data to answer specific research questions, rather than to develop a new theoretical framework, which is more in line with grounded theory. We have added a brief explanation in the Methods section explaining why we chose thematic analysis rather than grounded theory to assist readers who are unfamiliar with these terms. Please see lines 146-148 of the revised manuscript.

5. Lines 174-175 - inclusion criteria - self reported LBP, any LBP with or without radicular symptoms? This to me will make a difference when you are studying reassurance. Please clarify.

Response: Thank you for your comment. Our study aimed at people with low back pain. However, it is possible that people with different clinical presentations, such as radicular symptoms, may require a different type of reassurance. Based on the reviewer's comment, we have noted this as a limitation in the Discussion. Please refer to lines 390-392 of the revised manuscript.

6. I did not see a discussion of sample size for this survey. Can the authors provide some guidance or thoughts on this and if this was considered to be able to provide a response rate?

Response: Thank you for your comment. We targeted a sample size of 500 participants, which was calculated using the 95% margin of error formula with an estimated margin of error of 4.5%. We have added clarification in the Methods section. Please see lines 183-185 of the revised manuscript.

7. Statistics are solid and clearly articulated with the objective identified for each analysis run.

Response: Thank you for your comment and positive feedback regarding the clarity and appropriateness of our statistical analysis.

8. Excellent summary of the results with appropriate tables and figures.

Response: Thank you for your comment and positive feedback on the summary of results, tables, and figures.

9. Lines 285-287 - I think this first sentence needs to have stronger language as it is critical to start the discussion. The phrase "shed light" while appropriate is not strong enough language. Please revise. I say this because the next sentence has the evidence to back up stronger language.

Response: Thank you for your comment. We have revised the opening sentence of the Discussion to use stronger language, as suggested. Please see lines 301–302 of the revised manuscript.

10. In the limitations section, I would ask the authors to consider that in this study it appears that any back pain was include whether their was radicular symptoms or not. I think this is a limitation as different reassuring strategies may be needed with different types of LBP

Response: Thank you for your comment. We have added this point as a limitation in the Discussion section. Please see lines 390-392 of the revised manuscript.

11. Overall, I would like to thank the authors for this important manuscript that addresses a critical area of management in individuals with LBP.

Response: Thank you for your comments and for your positive feedback on our manuscript. We appreciate your thoughtful review.

---

## [Decision Letter · Decision Letter 1]

31 Jul 2025

The perception of individuals with low back pain regarding reassuring information: insights based on physiotherapists messages

PONE-D-25-18786R1

Dear Dr. Springer,

We’re pleased to inform you that your manuscript has been judged scientifically suitable for publication and will be formally accepted for publication once it meets all outstanding technical requirements.

Kind regards,

Nicola Diviani

Academic Editor

PLOS ONE

Additional Editor Comments (optional):

Reviewers' comments:

Reviewer's Responses to Questions

**Comments to the Author**

Reviewer #1: All comments have been addressed

2. Is the manuscript technically sound, and do the data support the conclusions?

Reviewer #1: Yes

3. Has the statistical analysis been performed appropriately and rigorously?

Reviewer #1: Yes

4. Have the authors made all data underlying the findings in their manuscript fully available?

Reviewer #1: Yes

5. Is the manuscript presented in an intelligible fashion and written in standard English?

Reviewer #1: Yes

Reviewer #1: Thank you for addressing my comments. I have no further suggestions for this paper. I want to thank the authors for their work.

**Do you want your identity to be public for this peer review?** For information about this choice, including consent withdrawal, please see our Privacy Policy

Reviewer #1: No

---

## [Editor Report · Acceptance letter]

PONE-D-25-18786R1

PLOS ONE

Dear Dr. Springer,

I'm pleased to inform you that your manuscript has been deemed suitable for publication in PLOS ONE. Congratulations! Your manuscript is now being handed over to our production team.

Kind regards,

on behalf of

Dr. Nicola Diviani

Academic Editor

PLOS ONE